# Substance Use and Addictive Behavior in Spanish Adolescents in Secondary School

**DOI:** 10.3390/healthcare9020186

**Published:** 2021-02-09

**Authors:** Elena García-García, María-Lara Martínez-Gimeno, José Alberto Benítez-Andrades, Joselin Miranda-Gómez, Enrique de Dios Zapata-Cornejo, Gema Escobar-Aguilar

**Affiliations:** 1San Juan de Dios Foundation, San Rafael-Nebrija Health Sciences Center, Nebrija University, 28036 Madrid, Spain; egarciga@nebrija.es (E.G.-G.); mmartinezgi@nebrija.es (M.-L.M.-G.); gescobar@nebrija.es (G.E.-A.); 2SALBIS Research Group, Faculty of Health Sciences, University of Leon, 27071 Leon, Spain; 3SALBIS Research Group, Department of Electric, Systems and Automatics Engineering, University of León, 24071 León, Spain; 4Center Psicología Clínica, 28010 Madrid, Spain; joselinmiranda@centerpsicologia.com; 5Sopra Steria, 28050 Madrid, Spain; enrique.zapata@soprasteria.com

**Keywords:** risk, addictive behavior, social environment, prevalence, internet, drugs

## Abstract

The detection and prevention of addictive behaviour at an early age is essential given the relationship between the age of the onset of consumption and the appearance of addiction disorders. The aim of this study was to describe the behavior related to substance use and addictive behaviors in adolescents at secondary school from 12 to 16 years of age. A cross-sectional descriptive study has been conducted. The prevalence of consumption of different addictive substances (alcohol, tobacco, cannabis, cocaine) and addictive behaviours (use of social networks and video games) were collated, and the influence of the surrounding social environment and risk perception were evaluated. The final sample was 1298 students. Alcohol, tobacco and cannabis use reflect the prevalence of last month’s consumption: 14% (11.8–15.6), 15% (13.4–17.4) and 3% (1.9–2.7) respectively. 76% of the sample frequently use the Internet (5–7 days per week). There is a positive association between the frequency of use and use in the immediate environment. The relationships found show the need for educational and preventive intervention aimed at parents and students that will allow them to know and effectively deal with possible problems associated with the consumption of addictive substances.

## 1. Introduction

The World Drug Report 2019 of the United Nations Office on Drugs and Crime (UNODC) reveals that at least 5.5% of the world’s adult population (271 million people) had used drugs in the previous year [1]. In relation to the damage caused by drug use, it is estimated that globally, in 2017, 42 million years of "healthy" life (Disability Adjusted Life Years, DALYs) were lost as a result of the use of drugs. In general, over the last 10 years, there has been a progressive increase in DALYs attributable to morbidity and mortality from all causes related to drug abuse [2].

Adolescence is a particularly vulnerable period for risky behavior related to alcohol and drug consumption, as well as for the development of addictions to other substances, such as Internet use and gambling. It is also a critical period for the adoption of healthy lifestyles and educational choices that play a determining role in adulthood [3,4].

Global figures inform us that the average age for the onset of consumption of the most prevalent substances, such as alcohol and tobacco, is between 13 and 14, although 8.50% of students start drinking alcohol and taking illegal and medical drugs between the ages of 12 and 14 (Ketamine, hallucinogens, inhalants at 12; amphetamines, San Pedro, Crack and others at 13; and tranquillisers and stimulants at 12 and 9) [1]. This coincides with other studies that report the onset of substance use such as tobacco, alcohol and cannabis in pre-adolescence (9–13 years), particularly in Western countries [5]. Likewise, gambling problems often become fully apparent in late adolescence or early adulthood. The prevalence of these problems in adolescents is between 4–8%, and they are considerably higher than in the general adult population, which is between 1–3% [4].

Focusing studies on Europe, the European School Survey Project on Alcohol and Other Drugs (ESPAD) collect data on substance use and other forms of risky behavior among 15 to 16 years-old students in order to monitor trends within, as well as between, countries. The results of this study report that cigarettes are one of the most easily accessible substances, with about 60% of the students reporting that it would be “very easy” for them to get hold of them if they wanted to. More than 18% ESPAD students had smoked cigarettes at age 13 or younger. Likewise, alcoholic beverages are perceived to be easy to obtain compared with other substances by almost 80% of students. In Denmark, Germany and Greece, this percentage rises to more than 90%. Over one third of these students (33%) had first tried an alcoholic drink at age 13 or younger, and 6.7% of students had experienced alcohol intoxication at age 13 or younger. In this study, cannabis is perceived to be the easiest illicit substance to get hold of, according to 32% of ESPAD students. On average, 2.4% of this population reported having used cannabis for the first time at age 13 or younger [6].

In Spain, the current prevalence of occasional drug use is 93.5% for alcohol, 72.5% for tobacco, 18.7% for hypnosedant, 9.5% for cannabis and cocaine and 5% for ecstasy, amphetamines and hallucinogens [7]. On the other hand, the figures for the occasional use of video games are around 70% for the general population, of which 9–23% play daily [8]. According to data from the Spanish Observatory for Drugs and Addictions (2020), 77.9% of secondary school students (14–18 years) have consumed alcoholic beverages on some occasion, 75.9% in the last year and 58.5% in the last month. With regards to tobacco, 41.3% have smoked at some time in their lives, 35% in the last year and 26.7% in the 30 days prior to the survey; and one third of young people smoke daily. Cannabis is the third most consumed drug among students between 14 and 18 years old, with 3 out of 10 students admitting to having used it at some time and 19.3% in the last 30 days [9].

In terms of influential factors in the initiation of substance use, family, socio-economic, cognitive and emotional aspects stand out, with significant social pressure and attitudes being a factor both in favor and against [2]. The Social Learning Theory contemplates the effects of association networks, through which it is explained that people with whom one interacts regularly establish behavior patterns that, on the other hand, when observed repeatedly, are more susceptible to being learned [10]. At a social level, the high degree of tolerance that exists towards these behaviors, as well as the low perception of risk associated with the consumption of toxic substances, have contributed to the generalization of consumption and the normalization of behavior. The perception of risk of drug-taking behavior, measured as the percentage of subjects who consider that this behavior may or may not cause problems, is an indirect indicator in the evolution of consumption [2].

With regards to gambling problems, studies suggest that a pattern of play behavior often occurs at an early age of 9–10 years, where parents, teachers and mental health professionals do not perceive children’s play as a serious health problem. Although most young people who gamble are often described as social, recreational or casual gamblers, a small but identifiable number of adolescents develop a serious gambling problem. Likewise, school-aged children are much more likely to participate in peer play activities (often related to games of skill), lottery and sports gambling; and as they grow older and have greater access to money and credit cards, they may begin to participate in the lottery, casinos and online gambling [11].

According to several studies, risk factors associated with gambling include male gender, age (18–25), low socioeconomic status, lower educational level, separated parents, early exposure to gambling, poor school grades, delinquency or history of truancy, alcohol and drug use, and impulsive behavior [4]. Protective factors include family cohesion, connection to school, motivation for achievement and effective coping skills [12].

The detection and prevention of addictive behavior at an early age is essential given the relationship between the age of the onset of the consumption and the appearance of addictive disorders [13,14]. In fact, regular users usually start between the ages of 11 and 13 with tobacco, alcohol, cannabis and cocaine [15]; In this area, a report from the Childhood Observatory of the Andalusian Regional Government indicates that a considerable proportion of the population begins to consume between the ages of 11–14 (alcohol 14.9%, tobacco 28.6% and cannabis 12.6%). Furthermore, among users of other drugs a high percentage begin to use them between the ages of 11–14 (cocaine 5.2%, cocaine base or crack 7%, MDMA (ecstasy) 3.4%, hallucinogens 2.6%, amphetamines or speed 4%, tranquillisers without a doctor’s prescription 6.5%, inhalants 22%, heroin 11%) [16]. Although the main studies into adolescent consumption tend to mark 14 as the starting age [9], or sometimes 15 [7], some studies, both national [16] and international [1], have reduced this range to 11–12, or even earlier.

On the other hand, the use of new technologies, social networks, video games and online games has increased in recent years [17]. This increase is associated with abusive consumption that can generate affective, psychosocial, mood or social disturbances [18]. These behavioral characteristics are very similar to those produced by drugs and highlight the addictive quality of these practices [19]. It is for this reason that it is especially important to study addictive behavior in adolescents, including all of the aforementioned psychoactive substances, as well as new technologies and video games.

Early detection of alcohol and substance abuse, as well as online leisure, is crucial for the implementation of prevention programs and consumption reduction policies. For this, it is necessary to know the behavior patterns of adolescents, the age of the onset of behavior adoption, and the social and family factors that may be related to acquired behaviors. Prevention among young people and students is a basic need for a public health approach. On the other hand, school is recognised as the ideal framework for implementing intervention programs aimed at reducing or at least delaying the start of addictive behaviors among adolescents, due to its universal scope. However, the problem lies in the fact that most of these programs are not evaluated, and those that are do not demonstrate a verifiable effectiveness in reducing and/or delaying addiction. There is therefore an international need to improve the implementation of prevention programs based on the best available evidence and culturally adapt them to students [20].

In the case of minors who use addictive substances like alcohol despite negative consequences, it is important to know not only the prevalence but also the behavior patterns. Most studies published to date focus on the amount consumed without taking into account leisure and free time styles [21]. Since the age of onset of addictive behaviors seems to be established below the age of 14 years, referred to in the National Survey of Consumption Habits (ESTUDES, “survey on drug use in Secondary Schools in Spain”), it is proposed to analyse leisure and consumption styles in adolescents from the age of 12 onwards, in order to establish the basis for the design of future prevention and control interventions.

The main objective was to describe the behavior related to substance use and addictive behaviors in Spanish adolescents at Secondary School, who attended 15 schools nationwide during the year 2019.

## 2. Methodology

### 2.1. Design

Prospective Transversal Descriptive Observational Study.

### 2.2. Participants

The reference sample framework for the selection of the sample was the population of students enrolled in the 1st to 4th years of secondary school in a network of state schools established at a national level in Spain. These schools have a homogeneous educational model and are located in different Spanish towns, with representation from all over the country. 

Inclusion criteria: Students enrolled in the 1st, 2nd, 3rd and 4th years of secondary school, who agreed to participate in the study.

Exclusion criteria: Students who are at home due to illness at the time of study.

Sample size and type of sampling: For the calculation of the sample size, a population estimate was made for a proportion based on a finite population of 3700 students. Assuming a 95% confidence level and an accuracy of 3 percentage units, a sample of 1080 students is required if 50% of the students initiate alcohol and illegal and medical drug use between the ages of 12–14. A 30% loss was calculated.

Consecutive non-probabilistic sampling was conducted, and the questionnaire was provided to all students present in the classroom who met the criteria for inclusion and exclusion.

### 2.3. Description of the Sample

Information was collected from 15 schools, with a minimum of 16 respondents and a maximum of 204 per school.

The final sample was 1298 students (35% of the accessible population) distributed homogeneously among the courses (Table 1), of which 53.6% (696) were male and whose ages range between 11 to 18 years old.

### 2.4. Variables and Measurements

The variables were defined though an adaptation of the ESTUDES survey, a survey carried out by the National Drug Plan in Spain. Similar surveys have been carried out in Europe and the United States (ESPAD study), which allows for comparisons. The ESTUDES survey has been carried out in Spain every two years since 1994, with the aim of determining the situation and trends in drug use and other addictions among students aged 14 to 18 in Secondary Education. It is financed and promoted by the Government Delegation for the National Plan on Drugs in this country [9].

The final questionnaire includes a total of 140 items, written in Spanish, distributed as follows:

Sociodemographic information is collected from the participant: sex, age, family and cohabitation status, country of birth, as well as the country of birth and employment status of the parents (Table 1).

Variables related to academic performance and leisure habits were collected, such as self-consideration as students, usual qualifications, self-perception of academic performance (Table 1), leisure activities in the last 12 months and an estimate of the money they have for their expenses.

With regards to drug taking, information was collected on the occurrence of substance use (alcohol, tobacco, caffeine, cannabis, cocaine, ecstasy, hallucinogens, drugs, opiates, inhalants, liquid ecstasy and steroids) at some time in one’s life, in the last year, and in the last month.

Information was also collected on the frequency of participating in behaviors such as surfing the Internet, offline and online gaming, use of instant messaging platforms, social networks and virtual reality (VR)-type video games and multiplayer online role-playing games massive (MMORPG) in the last year.

With regards to perceived consumption risk, the questionnaire includes questions that refer to the risk that the student associates with each type of behavior. For example: “We would like to know your opinion about the problems (health or otherwise) that each of the following behaviors may involve”.

Consumption and leisure behaviors in the Surrounding Social Environment (father, mother, siblings and friends) of the students were also taken into account.

### 2.5. Data Collection

The project was approved by the management and legal office of the participating schools, as well as by the Research Commission of the San Juan de Dios Foundation (FSJD) and the Clinical Research Ethics Committee of the Niño Jesús University Children’s Hospital in Madrid, with the internal code R-0009/19, on 26 February 2019. The schools were first contacted through the San Juan de Dios Foundation, and then the teachers in charge of the groups of participating students were informed about the objectives of the study and the procedure for collecting data. A person responsible for data collection (someone not responsible for the groups and who did not teach or had not taught them) was available in each center.

Secondly, all parents whose children were eligible for the study were contacted and sent the information sheet, requesting informed consent. They were informed that at the end of the study no information on the individual data of the participants would be given, but a report with the overall results would be made available.

Data were collected through a completely anonymised online questionnaire, so that once the questionnaire was sent, the individuals could not be identified, thus ensuring their anonymity. This process was carried out in the school in the time designated for “prevention tutorials” established in the school timetable of each center, remaining in the classroom throughout the process and ensuring that the questionnaire was completed individually. The information was collected using an electronic tablet through which the students had access to the survey link. All students could decide anonymously whether or not to participate in the study by sending in the questionnaire.

Reference is made to the convenience of the teacher not being in the classroom during the application, as his or her presence could cause students to distrust the anonymity of their answers. If the teacher is present, it is indicated that he or she should not walk around the classroom, nor should he or she explain contents or address the students during the completion of the questionnaire.

During the survey period, the interviewers made sure that the questionnaire was filled in individually.

In order for the study subjects to identify certain terms and substances well, colloquial names were used for the substances, in the same way that occurs in surveys at European and national levels.

### 2.6. Data Analysis

The frequency of consumption by the sample was estimated. An estimate was also made of the frequencies in the last year of behaviors such as surfing the Internet, using instant messaging platforms, social networks, and VR and MMORPG games. Using Pearson’s Chi-square test, the association between the occurrence of consumption by the subject in the last month (alcohol, tobacco, cannabis and/or cocaine) and the daily consumption of those same substances by their close social environment (father, mother, siblings, friends and/or all the above) was evaluated. Likewise, the relationship between the frequency of use of social networks and videogames by the subject and the intensive use (7 h or more a day) of surfing the Internet and/or playing videogames by their close environment was evaluated. Pearson’s Chi-square test was conducted separately for each addictive behavior and for each member of the immediate environment.

Furthermore, the association between the subjective perception of risk associated with different consumption activities and the frequency of abuse of those substances by the subject was also evaluated using Pearson’s Chi-square test. Pearson’s Chi-square test was used for each addictive behavior separately.

Due to high sensitivity to sample size when using the Chi-square test, in all cases we considered a 99% confidence level, and the results were considered significant with an associated probability value of *p* < 0.01.

## 3. Results

### 3.1. Students’ Substance Use and Leisure Habits

In terms of alcohol, tobacco and cannabis consumption, the results show that 13.7% (1.8–15.6), 15.4% (13.4–17.4) and 2.8% (1.9–2.7), respectively, were consumed in the last month. Lifetime use of other drugs stands at 1.3% (17) for cocaine, 3.3% (43) for opiates and inhalants, and values between 0.2% (2) and 0.8% (11) for steroids, ecstasy or hallucinogens. Frequencies of use are shown in Table 2.

In most substances, consumption shows an association with the year, with higher consumption in higher courses. Alcohol and tobacco consumption are higher as the course increases, showing statistically significant differences (*p* < 0.001; *p* = 0.003, respectively) (Table 3).

As regards the Internet and social network use, 91.1% (1182) make frequent (more than 5 days/week) or very frequent (2–4 days/week) use of messaging (WhatsApp, WhatsApp Ireland Limited, Dublin, Ireland); 86.7% (1125) use social networks; 75.5% (980) watch films, listen to music or watch series; and 35.7% (463) play virtual reality games. Statistically significant differences were found in the use of social networks, being lower in 1st year (*p* < 0.001).

### 3.2. Students’ Addictive Behaviors Related to Surrounding Social Context Addictive Behaviors

In assessing the relationship between students’ addictive behaviors and their social environment, we found that the addictive behaviors adopted by friends are statistically significantly related to students’ behavior in all the variables analyzed (alcohol, tobacco, and drug use, and use of the Internet, social networks, and video games). Similarly, sibling behaviors related to Internet and video game use are significantly related to student behaviors. However, if we focus on substance use, this relationship only appears significant for cannabis. Parents seem to influence cannabis use as well. The relationships between students’ addictive behavior and the behavior of their surrounding social environment are shown in Table 4.

### 3.3. Risk Perception and Frequency of Adopting a Risk Behaviour

With regards to alcohol, no relationship was found between risk perception and drinking 5–6 glasses every weekend and getting drunk regularly every time (*p* = 0.026). Drinking 5–6 glasses in less than 2 h (binge-drinking) also did not present an association (*p* = 0.032).

No association was found when assessing the relationship between perceived risk of smoking and tobacco use. However, a statistically significant relationship was found with a low perceived risk of regular marijuana or hashish (cannabis) smoking and increased frequency of cannabis use (*p* < 0.001), as well as with use within a maximum time period of 2 h (*p* < 0.001). No significant association was found in relation to low perceived risk of regular cocaine use and the increased frequency of cocaine use (*p* = 0.032).

No statistically significant relationship was found with the low perceived risk of using the internet and video games until the early hours of the morning on a regular basis with the frequency of Internet use (*p* = 0.301), nor with the frequency of wanting to stop the connection and not being able to, (*p* = 0.215), as shown in Table 5.

## 4. Discussion

The main objective of the study was to describe behavior with regards to substance use and addictive behaviors in Spanish adolescents at Secondary School, aged between 12 and 16 years, who attend 15 schools nationwide—a subject widely studied in the literature from the age of 14 [7,9,22].

In Spain, the results of a national study into consumption habits conclude that alcohol is the most consumed psychoactive substance among Secondary School students aged 14 to 18, with 78% having consumed it at some time in their lives and 59% in the last month. The second most prevalent drug found was tobacco, with 41% having smoked in their lives and 27% in the 30 days prior to the survey. Cannabis is the third most widespread drug among students in this age group, and it is also the most prevalent illegal substance, with 33% of students using it occasionally, 28% saying they had used it in the past year, and 20% in the past 30 days. With regards to hypnosedants, with or without a prescription, they are the fourth most prevalent substance, with 18% of students having taken this type of substance at some time in their lives. Cocaine is reported by 3% of students who have used it at some time in their lives [9].

In our sample, the data in relation to alcohol are lower, although they maintain the same trend of more frequently consumed substances [9], with 31% of students stating that they have consumed alcohol at some time in their lives and 14% in the last 30 days. Tobacco is also the second most prevalent drug, with consumption at 15% and 9%, respectively. Cannabis remains the third most prevalent drug, with data of 7% and 3%, respectively. Cocaine is reported by 1% of students, who have used it at some point in their lives. This percentage difference found with respect to other studies [9,22] can be related to the age of the participants, as this study includes three additional age groups (11, 12 and 13 years of age), taking into account that one of them (13 years) represents 23% (304 students) of the total sample. This hypothesis can be confirmed with the results reported by the Andalusian Childhood Observatory (2017), which showed alcohol consumption figures of around 26% in the last month, increasing to 50% when the data were reported by students between 16 and 20. This report concludes that consumption is mainly concentrated at the weekends, with 3% in the 12–13 age group, 22% in the 14–15 age group and 50% in the 16–20 age group [23]. This trend of increasing consumption at a later age has also been found in the present study.

The report of the United Nations Office on Drugs and Crime (2013) confirms that alcohol is the legal substance most consumed by schoolchildren, with the average age of initiation being 13 and the lowest age for the onset of consumption being 8. The study shows that 25% of schoolchildren between the ages of 8 and 11 have consumed alcohol at some point in their lives; however, 50% of schoolchildren started drinking when they were between 12 and 14. With regards to tobacco, data from this study show that 23% of adolescents consumed tobacco or alcohol in the last year and 12% in the last month, similar figures to those found in our study [24]. These conclusions led us to initiate monitoring studies of drug consumption at ages below 14 years of age in order to detect the real age of initiation and establish appropriate prevention policies. On the other hand, studies such as the one conducted by Peltzer et al. (2015), which analyzed the correlations between early initiation (<12 years) in the consumption of cigarettes (prevalence of 16% among 6540 participants), alcohol (14%) and cannabis (13%) with the prevalence of suicidal ideation and behavior in school-aged adolescents (26%), detected a strong association when the age of consumption occurs in pre-adolescence. It was further concluded that the combined early initiation of two or three substances was highly associated with suicide attempts, not just with ideation [5].

These conclusions led us to initiate monitoring studies of drug consumption at ages below 14 in order to detect the real age of initiation and establish appropriate prevention policies. The analysis of the influence of the respondents’ environment in this study shows a pattern in which, if parents smoke, so do their children—a relationship that is stronger in the group of students who report frequent tobacco use (3–4 times a week).

In the exploratory analysis of the profile of students who abuse substances (alcohol, tobacco and other drugs) and who abuse the use of online gambling/video game channels, it is observed that the profiles of video game abuse present weak relationships with the consumption profiles of certain substances, and that these relationships, when they do present themselves, are inversely proportional: the greater the degree of addiction to video games, the lesser the degree of addiction to substance consumption. The same happens when studying the profiles of online gambling and substance consumption. However, since few responses were available with a profile of online gambling abuse, these results are inconclusive, and if this inverse relationship is maintained in future studies, it should be explored.

With regards to the perception of risk, according to the OEDA (2018), the perception of people who partake often decreases significantly compared with people who do not partake, and they are also more likely to consume more than one substance at a time. Data from several surveys [9,22] show an increasing evolution over time in terms of perceived risk of consumption and prevalence. In the present study, the data indicate that there is a relationship between a lower perception of risk and a higher frequency in the consumption of cannabis. No relationships were found for the other comparison, although the perception of risk in the consumption of tobacco is very low. This informs us that the sample analysed uses tobacco independently of the perceived risk, which could be due to the impulsive personality traits associated with tobacco and alcohol users preventing them from contemplating the consequences of consumption and their risks at the time of consumption [25].

With regards to the relationship between consumption and risk perception, it can be seen that respondents who consume alcohol are aware of the risk involved in their consumption, perceiving more risk than those who do not. However, with regard to tobacco, those who consume perceive less risk than those who do not. Those who use cannabis, on the other hand, perceive less risk than those who do not, a hypothesis that is consistent with the results of Herruzo et al. (2019), who conclude that cannabis users are grouped into three groups: those considered ‘strict’ (53%) with a perception that use ‘always causes problems’, the ‘permissive-conscious’ (31%), and the remaining 15% who are considered ‘lax’ participants, with responses such as cannabis “rarely causes problems” [26]. The data led to the conclusion that cannabis consumption during the previous 30 days was 14 times more frequent among young people in the cluster with a lax perception of cannabis-related risk than among those in the cluster with stricter risk perception [26].

The data obtained in this study shows the importance of the family and the immediate environment in the image that adolescents have of the consumption of substances, but also of addictive behaviors such as the Internet, social networks and video games, with the latter being the least treated and informed in school and family environments [22]. According to Gallegos et al., emotional detachment from parents and peer pressure promote consumption at high school ages among those who have previously started using alcohol and tobacco [27].

The communication and information that adolescents receive is vital for them to have a real and reliable perception of risk in relation to addictive behaviors and to reduce their frequency of consumption. Furthermore, if information is provided by educational centers, it will reach all students equally, forming the close social circle of each one and helping to modulate consumption in the social circle. At a national level, it is considered interesting to implement preventive campaigns that cover addictive behaviors beyond tobacco, alcohol, cannabis and cocaine, also considering the dangers of abusive use of the Internet and video games without control or self-regulation, which has a relationship to depression and attention deficit disorders, as well as being related to insomnia, failure in learning processes, low academic performance and dropping out of school [28].

In general, it is observed that there is a great variety of consumption profiles in the environment explored, suggesting similar behaviors among equals (parents-mothers, friends-brothers), although the correlation does not mean causality, and it would be interesting to explore it with other types of studies.

Based on the prevalence and risk perception results detected in the study, it is recommended that schools, specifically school nurses, approach prevention talks not in terms of risks but in terms of the conscious decision-making process, since the long-term consequences that addictive behaviors can have are not perceptible at that age. Involving students in their decisions and in the design of prevention strategies can promote empowerment.

Finally, we would like to highlight the possible limitations of the study. The participating schools belonged to the same organisational group, with concerted and religious links, so the results may not be extrapolated to the general population. On the other hand, using the ESTUDES survey [22] as a basis for the design of the questionnaire, which was designed for students over 14 years of age, has made it possible to compare the results, but this may also have generated a limitation due to the understanding of the content by younger students.

Other limitations to be highlighted, which would have allowed the correlation of data and extrapolation to social contexts, were the lack of data collection related to the socio-economic context of the region in which each school is located, as well as the local tradition in consumption habits. In this sense, cross-checking the data with epidemiological databases of alcohol, tobacco and substance consumption at the adult population level would have been of interest.

The necessary authorization from parents for the participation of adolescents in the study is considered a limitation because they do not want to provide data on consumption habits and behavior of the family unit, despite guaranteeing the anonymity of the responses. This situation may explain the differences found in the levels of participation of the different schools.

## 5. Conclusions

The conclusions drawn from this study can be mainly summarised into four areas:Regarding the consumption of alcohol, tobacco and cannabis, the behavior pattern shows an association with the school year, with higher consumption occurring in higher years.Higher consumption of alcohol and tobacco is related to higher consumption in the surrounding social context (friends and siblings, respectively).Regarding the pattern of Internet use and social networks, a positive relationship is established between the frequency of consumption and consumption by friends and siblings.The results of the study show the prevalence data on addictive behaviors, substances and online or Internet gambling in minors from 11 years of age onwards.In relation to nursing practice, actions linked to adolescent participation in decision making and awareness of the process are recommended. Alternatively, health education programmes that focus on highlighting the risks of addictive consumption of substances or abusive use of technology are not recommended since, as the results of this study show, knowing the risks does not imply not consuming.For future lines of research, we propose broadening the age range of national studies on consumption from the current 11 to 14 years of age, as this approach to accessing illegal substances and hypnosedative drugs is of interest. It should be noted that the latter are subject to medical prescription, so they cannot be freely purchased at the pharmacy.The established prevention programmes, both in schools and at a community level, seemingly do not have a strong impact on consumption figures. Therefore, as long as prevalence remains constant over time [9], we propose changing the design approach of interventions, betting on participatory studies in which the collective (of adolescents) is empowered, allowing them to participate in the design of interventions that promote healthy leisure in the face of addictive behaviors, from a conscious health perspective.

## Figures and Tables

**Table 1 healthcare-09-00186-t001:** Description of the sample of the study.

**N = 1298 *; *n* (%)**						
Age Average (SD)				13.95 (SD 1.26)		
Sex		Man		604 (46.38)		
		Woman		696 (53.62)		
Secondary school grade			1st year	2nd year	3rd year	4th year
			319 (24.6)	359 (27.6)	340 (26.2%)	280 (21.6%)
Nationality		Spanish	Other European	Latin America	Asia	Africa
	Sample	1221 (94.07)	13 (1)	46 (3.54)	14 (1.08)	3 (0.23)
	Mother	1090 (83.98)	46 (3.54)	135 (10.4)	22 (1.69)	0 (0.00)
	Father	1106 (85.21)	46 (3.54)	106 (8.17)	18 (1.39)	11 (0.85)
Cohabiting		Mother	Father	Both (together)	Both (separately)	Others
		151 (11.63)	16 (1.23)	990 (76.27)	89 (6.86)	46 (3.54)
Mother’s job		Housework	Outside the home	Unemployed	Retired/pensioner	Deceased
		176 (13.56)	1024 (78.89)	63 (4.85)	8 (0.62)	9 (0.69)
Father’s job		Housework	Outside the home	Unemployed	Retired/pensioner	Deceased
		7 (0.54)	1156 (89.06)	36 (2.77)	39 (3)	18 (1.39)
Self-consideration as a student		Very bad	Bad	Normal	Good	Very good
		16 (1.23)	75 (5.78)	496 (38.21)	505 (38.91)	206 (15.87)
Usual qualifications		F	E-D	C	B	A
		90 (6.93)	211 (16.26)	246 (18.95)	492 (37.9)	259 (19.95)
Self-perception of academic performance		I should have worse scores than those which I have				66 (5.08)
		I have the scores I deserve				774 (59.63)
		I should have better scores than those which I have				456 (35.13)
Disposable money per month		€0	Up to €10	From 10 to €50	>€50	As needed
		215 (16.56)	646 (49.77)	294 (22.65)	51 (3.93)	26 (2.00)

* Data “do not know/no answer” not included, frequencies in the different variables oscillate between 0.08–2.93; SD: Standard Deviation; The Spanish educational system qualifies based on 10 points where: F < 5; E–D = 5–5.9; C = 6–6.9; B = 7–8.9; A = 9–10.

**Table 2 healthcare-09-00186-t002:** Frequencies of consumption of substances.

Substance	During All Life	Last Year	Last Month
Alcohol	399 (30.7%)	358 (27.6%)	178 (13.7%)
Tobacco	200 (15.4%)	168 (12.9%)	110 (8.5%)
Cannabis	91 (7%)	70 (5.4%)	36 (2.8%)
Cocaine	17 (1.3%)	12 (0.9%)	4 (0.3%)
Ecstasy	10 (0.8%)	6 (0.5%)	3 (0.2%)
Hallucinogens	15 (1.2%)	9 (0.7%)	5 (0.4%)
Opiates	43 (3.3%)	11 (0.8%)	5 (0.4%)
Inhalers	43 (3.3%)	14 (1.1%)	8 (0.6%)
Liquid ecstasy	4 (0.3%)	2 (0.2%)	2 (0.2%)
Steroids	3 (0.2%)	2 (0.2%)	2 (0.2%)

**Table 3 healthcare-09-00186-t003:** Differences by grade in substance use and leisure habits (N = 1298).

Grade	Alcohol	Tobacco	Cannabis	Cocaine	Internet *	VR Games	MMORPG
		Consumption per Month			Consumption +5 days/week		
1	15 (4.7%)	13 (4.1%)	7 (2.2%)	1 (0.3%)	151 (47.3%)	52 (16.3%)	11 (3.4%)
2	31 (8.7%)	28 (7.8%)	7 (2.0%)	3 (0.8%)	248 (69.3%)	64 (17.9%)	13 (3.6%)
3	55 (16.2%)	36 (10.6%)	9 (2.6%)	-	257 (75.6%)	56 (16.5%)	17 (5%)
4	77 (27.5%)	33 (11.8%)	13(4.6%)	-	236 (83.9%)	41 (14.6%)	6 (2.1%)
Total	178 (13.7%)	110 (8.5%)	36 (2.8%)	4 (0.3%)	892 (68.7%)	213 (16.4%)	47 (3.6%)
*X* ^2^	76.29	14.06	4.93	5.18	120.45	14.09	10.56
*p*	<0.001	0.003	0.177	0.159	<0.001	0.295	0.567

* Use of social Networks or Internet; VR: Virtual reality; MMORPG: Massive Multiplayer Online Role-Playing Game; *X*^2^: Chi square; *p*: *p* value.

**Table 4 healthcare-09-00186-t004:** Relations between students’ addictive behavior in the last month and surrounding social environment behavior (N = 1298).

Social Environment Behavior	Alcohol	Tobacco	Cannabis	Cocaine	Internet *	VR Games	MMORPG
	X^2^; *p* value						
Friends	9.703; *p* = 0.002	48.310; *p* < 0.001	168.396; *p* < 0.001	9.919; *p* = 0.002	17.670; *p* = 0.001	49.524; *p* < 0.001	31.769; *p* < 0.001
Mother	2.530; *p* = 0.112	7.875; *p* = 0.005	33.186; *p* < 0.001	0.006; *p* = 0.937	3.060; *p* = 0.548	5.942; *p* = 0.204	1.358; *p* = 0.851
Father	1.169; *p* = 0.280	1.261; *p* = 0.261	17.370; *p* < 0.001	0.044; *p* = 0.834	0.752; *p* = 0.945	10.980; *p* = 0.027	15.750; *p* = 0.003
Siblings	1.169; *p* = 0.280	6.642; *p* = 0.010	27.703; *p* < 0.001	0.003; *p =* 0.956	19.482; *p* = 0.001	24.653; *p* < 0.001	18.713; *p* < 0.001
All Together	0.044; *p* = 0.833	15.974; *p* < 0.001	5.524; *p* = 0.019	0.012; *p* = 0.911.	9.199; *p* = 0.056	13.509; *p* = 0.009	7.960; *p* = 0.093

* Use of social Networks or Internet > 7 h per day; VR Games: Virtual reality games > 7 h per day; MMORPG: Massive Multiplayer Online Role-Playing Game >7 h per day; X^2^: Chi square; *p*: *p* value.

**Table 5 healthcare-09-00186-t005:** Perception of risk consumption.

Consumption	Perception of Risk	
	X^2^	*p* Value
Alcohol (5–6 glasses every weekend)	17.424	*p* = 0.026
Alcohol (>5 glasses <2 h/ Binge drinking)	16.830	*p* = 0.032
Tobacco (>1 packet a day)	3.601	*p* = 0.731
Tobacco (>1/2 packets in two hours)	7.572	*p* = 0.271
Cannabis	65.707	*p* < 0.001
Cocaine	13.797	*p* = 0.032
Internet and videogames until the early hours of the morning	9.516	*p* = 0.301

X^2^: Chi square; *p*: *p* value.

## Data Availability

The data presented in this study are available on request from the corresponding author. The data are not publicly available due to privacy/ethical restrictions.

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
