# Peer review of "Substance Use and Addictive Behavior in Spanish Adolescents in Secondary School"

_healthcare, 2021, doi:10.3390/healthcare9020186_

Round 1

Reviewer 1 Report

The present manuscript entitled "Substance use and addictive behavior in Spanish adolescent in Secondary School" is an article that tries to describe behavior related to substance use and addictive behaviors in Spanish high school adolescents. It is an interesting manuscript, it presents certain minimum limitations that are detailed below:

- At the end of the introduction it closes with the aim of the study. The specific objectives are detailed below by points. It would not be necessary to detail the main objectives, it would be better to close with the objective of the study in general.

- There are some acronyms in the manuscript that are not developed, for example ESTUDES.

- On line 219, where the ethics committee section is mentioned, it would be better to detail the authorization number of the ethics committee and the date.

- At the end of the conclusions it would be convenient to add some future lines, how to put the results into practice.

- Some references do not comply with the journal regulations, for example reference 1.

Author Response

We have attached a document with the answers. Thank you very much.

Reviewer 2 Report

Dear authors.

I have now taken the time to review your great article. Please find below the complete review report., which includes only suggestions to improve your manuscript.

I wish you the best with your ongoing research.

Healthcare 1090873-v1. 22nd of January 2021.

This article by Garcia-Garcia et al is exploring substance use in relation to addictive behaviors in adolescent, performed throughout Spain. The manuscript will be extremely helpful for readers to put drug consumption in adolescents into perspective. Moreover, considering the relatively high prevalence of consumption, public health organisms and local policy makers should also find this article very helpful. The article is easy to read and easy to understand. Besides, the manuscript is well structured. The discussion is very nicely written. Therefore, I have only (very) minor suggestions to improve the draft of this manuscript. These are detailed line by line below.

Lines 33-34: I suggest revising “damage caused for the use of drugs…” into “damage caused by drug use…”

A full stop is missing at the end of the sentence on line 37.

Lines 39-40, authors should edit the sentence “to other substances, the Internet and gambling” into “to other substances, such as Internet use and gambling”.

A space is missing on line 48, between “countries” and the reference number 5.

Line 62, authors should consider revising “In this study the cannabis is…” into “In this study, cannabis is …”.

Lines 68-69. It is rather odd to use the term “consumption” when referring to ”video games”. Authors should maybe consider using “usage” or “use” in such a sentence.

Lines 77-85 are missing references. Authors should add specific references.

Line 97. Authors should consider editing “According to studies…” into “According to several studies…”.

Line 115. Again, authors should avoid using “consumption” when referring to “technologies, social networks, video games and online game”. Please consider editing, as mentioned in a previous comment.

Lines 120-121 has a typographical error. Please consider editing “including all the of the aforementioned…”. Into “including all of the aforementioned…”.

Line 134. A space is missing at the end of the sentence.

Line 200. What are “usual calcifications” ? Do authors mean “usual qualifications” ? Please consider explaining this.

In the Methods section, within part 2.3, can the authors please explicit what language was used in the questionnaires ? Readers would assume that the questions were written in Spanish. Is that accurate ?

Authors have explained data collection very well. I find it very re-assuring that authors put all efforts to retain anonymity of the students, as well as protecting them from biased answers. Congratulations, data collection and ethical considerations were performed at the state-of-the-art.

Lines 287-288. Again, authors should avoid using “consumption” when referring to “films, music or series […] virtual reality games”. Please see comments mentioned above.

Line 295 contains an odd spacing (double spacing maybe?). This is also seen on line 298.

Line 311 is missing the number of hours. Is it less than 2 hours ? (this is mentioned in Table 5, right ?).

Table 5 contains some typographical errors on the word “tobaco”. Authors should correct this into “tobacco”.

Line 342, a parenthesis is used once, but is never closed.

Line 394 contains a typographical error within “consumption in of cannabis”.

Line 396, do authors mean “consumption in of cannabis” or just “consumption of cannabis” ?

Lines 412-425. This is a very well written paragraph and very interesting. Can authors also briefly mention “mirroring behaviour” and “peer pressure” ? Indeed, countless articles are available regarding the link between peer pressure (friends) and drug consumption in adolescents. Besides, “mirroring” adult behaviour (drinking, smoking, etc…) is also widely studied amongst adolescents. Authors should consider briefly mentioning both of these traits.

It is afflicting to see such high percentages of consumption for very addictive substances (cocaine, opiates) in youngsters. Can the authors provide some information, in the discussion, on the following :

1/ How is the Spanish government protecting these children from accessing these (very) addictive substances? Could reference number 9 be used as guideline in Spain ?
2/ Is prevention put into place by the authorities (and is it accessible at/after school) ?
3/ How are the drugs acquired/purchased by these children? Do authors have any data on the link between the amount of money given to children and quantities of drug purchased ?

Finally, I suggest revising the English throughout the manuscript (for example the use of the term “in regards”, which is not use correctly). The team at MDPI will surely help you in doing so.

Author Response

(The authors gave the same response as above.)
